# Impact of Alternate Gait Training Using Knee–Ankle–Foot Orthoses with Oil Damper Ankle Hinge in Patients with Subacute Severe Hemiplegia

**DOI:** 10.3390/brainsci11111430

**Published:** 2021-10-28

**Authors:** Hiroaki Abe, Kazutaka Nishiyama, Yuichi Yamamoto, Toru Okanuka, Yasuhito Yonezawa, Koji Matsumoto

**Affiliations:** 1Department of Physical Therapy, Fukushima Medical University School of Health Sciences, 10-6, Sakaemachi, Fukushima 960-8516, Japan; 2Department of Rehabilitation, Kita-Fukushima Medical Center, 23-1 Aza-higashi, Hakozaki, Date, Fukushima 960-0502, Japan; penntagon0104@yahoo.co.jp (K.N.); reha_yuichi@jinsenkai.or.jp (Y.Y.); 3Department of Rehabilitation Medicine, Kohnan Hospital, 4-20-1, Nagamachi-Minami, Taihaku-ku, Sendai 982-8523, Miyagi, Japan; ookanuka@kohnan-sendai.or.jp; 4Pacific Supply Co, Ltd., 1-12-1, Goryou, Daito, Osaka 574-0064, Japan; yonezawa@p-supply.co.jp (Y.Y.); matsumt@p-supply.co.jp (K.M.)

**Keywords:** gait training, hemiplegia, knee–ankle–foot orthoses, oil dampers ankle hinge, assistive device

## Abstract

Patients with severe hemiplegia along with knee instability require knee–ankle–foot orthoses (KAFOs) for gait training. However, in these patients, it is unclear which type of walking training is more effective to improve gait function. Providing alternate gait training (AGT) improves walking function in patients with spinal cord injury, but it is still unclear whether this is effective in hemiplegic stroke patients. In this study, we defined “unified AGT” as AGT performed with the same therapeutic concept by physiotherapists. We then investigated whether AGT improved gait function quicker than our traditional gait training in hemiplegic stroke patients. We enrolled 15 subjects with severe hemiplegia and knee instability who had undergone unified AGT using KAFOs with hinged oil dampers at the ankles, and 30 historical control (HC) subjects who had undergone traditional gait training. We used multiple comparison and survival analyses to analyze the differences in the functional independence measure (FIM) gait score changes between the two groups. The multiple comparison revealed a significant increase (*p* < 0.05) in the FIM gait score compared with its initial score in the subjects with unified AGT. However, this improvement was not seen in the HC subjects. Additionally, the survival analysis of time taken to recover revealed a significant difference between the subjects with unified AGT and HC subjects (*p* < 0.05). These findings suggest that unified AGT using a KAFO facilitates gait improvement in patients with severe hemiplegia and knee instability.

## 1. Introduction

Regaining gait function is a major goal of stroke rehabilitation [1]. A previous study reported that, at the end of rehabilitation, 21% of stroke patients had died, 18% were unable to walk, 11% could walk with assistance, and 50% could walk independently [2]. It has also been reported that the time and extent of recovery are related to both the degree of initial impairment of walking function and the severity of lower leg paresis [2]. Nevertheless, gait function can be regained following sufficient gait training in some severe stroke patients [3]. The use of several leg orthoses, such as ankle–foot orthosis (AFOs) and knee–ankle–foot orthoses (KAFOs), for patients with hemiplegic stroke, promotes active rehabilitation and facilitates an expedient return home following the improvement in daily living activities [4] (see Appendix A). Moreover, patients with severe hemiplegia can benefit from using KAFOs because these patients exhibit both severe ankle and knee instabilities [5,6]. Studies have reported the effect of a KAFO in improving knee recurvatum [7], preventing severe knee bending, and improving stability [5,6] (for knee instability, see Appendix A).

Gait function can be influenced by the voluntary motor control system, such as the corticospinal tract, by the involuntary motor control system such as the mesencephalic-reticulospinal, and by the central pattern generator in the spinal cord [8,9,10].

In patients with spinal cord injury, alternate stepping movements with afferent input from “load receptors” induce a patterned leg muscle activation akin to those induced in healthy subjects [11,12]. These inputs induce lower limb muscle activity in patients who have difficulty in voluntary lower limb movement. Therefore, we assumed that providing an alternate gait pattern for patients with severe hemiplegia induces afferent load, and proprioceptive receptor inputs would induce earlier improvement in the gait function. Afferent information from the bilateral hip joints appears essential for the generation of locomotor-patterned leg muscle activation, but unilateral stepping movements lead to inadequate leg muscle activation [11,12]. Moreover, patients with severe hemiparesis show poor stability in the paretic side lower limb and difficulty in walking without lower limb support. To achieve good stability, strong external support is required, such as a KAFO. We speculated that gait training using a KAFO would facilitate gait function recovery in patients with severe hemiparesis. We defined gait training with alternate large hip flexion and extension as “alternate gait training (AGT)” using a KAFO with the oil damper ankle hinge, and also defined “unified AGT” as AGT performed with the same therapeutic concept and training techniques by trained physiotherapists (for AFOs, KAFOs and AGT, see Appendix A).

The purpose of the present study was to investigate whether unified AGT improves gait function quicker than our traditional gait training in hemiplegic stroke patients. In our traditional gait training, based on the subjective decision of each physiotherapist, even in cases with knee instability, the training was performed with AFOs instead of KAFOs. We hypothesized that the provision of afferent information from the bilateral hip joints by unified AGT would induce locomotor-patterned leg muscle activation and improve gait function early in hemiplegic stroke patients (see Appendix A).

## 2. Materials and Methods

### 2.1. Study Design

This is a case-matched historical control (HC) study.

In 2016, the institution where the study was conducted began unified AGT with a KAFO as part of routine therapy for appropriate patients with subacute stroke. This prevented us from conducting a randomized controlled trial, as it would not have been ethical to allocate subjects to a no-AGT control group when the intervention was part of routine care. Thus, we prospectively recruited individuals who were provided with unified AGT during inpatient rehabilitation and compared their gait ability with that of matched HC subjects for whom our traditional gait training was provided.

All procedures were approved by the local Ethics Committee of the Institutional Review Board (or Ethics Committee) of Kita-Fukushima Medical Center (protocol code 59, 26 November 2015 approval) (study-59) and were consistent with the Declaration of Helsinki. Informed consent was obtained from all subjects.

### 2.2. Subjects

#### 2.2.1. Prospective Unified AGT Subjects

Prospectively, subjects were recruited from stroke patients who had received inpatient rehabilitation with unified AGT at the Department of Rehabilitation, Kita-Fukushima Medical center, Japan, between March 2016 and April 2017. Subjects were eligible if they: (1) required assistance in gait training at admission, and were assessed with a gait score of 1 according to the functional independence measure (FIM) [13,14], and (2) needed a KAFO (Gait Innovation (GI), Pacific supply, Osaka, Japan) at least 2 weeks from the first gait training because of severe knee instability. Patients who needed a KAFO were identified based on the following conditions: (1) those who used an AFO but were unable to walk properly; (2) those who could barely walk with an AFO due to gait abnormality because of buckling knee or knee extension thrust, in the loading response phase; (3) those who did not have profound limitations in the range of motion of their legs to undergo unified AGT using a KAFO; (4) those who had moderate to severe motor paresis and a stroke impairment assessment set (SIAS) [15,16] motor knee score of ≤3 (a score of 3 means that the knee joint can be extended against gravity and with some clumsiness); (5) those who had rehabilitation motivation, or those whose vitality index (VI), [17] sub score of rehabilitation was over 0. Patients were excluded if the length of hospital stay (LOS) was less than 3 months as FIM could not be evaluated four times. A total of 15 subjects met the above criteria (Figure 1).

#### 2.2.2. HC Subjects

A matched sample of HC subjects was selected from 123 eligible individuals who had lower limb paresis (SIAS motor score of knee extension ≤3) in an HC from the same institution between October 2012 and March 2016. The HC subjects were matched based on age (range of 68–86 years), gait ability (initial FIM gait score = 1), motivation (VI > 0), and LOS. Thirty subjects met the above criteria (Figure 1).

Of the 30 HC patients, 12 used a KAFO, 11 used an AFO, and 7 did not use any brace during a gait training program. Among the 12 patients who used a KAFO, 8 achieved a transition from a KAFO to an AFO, known as “cut-down”, but 4 still required a KAFO at discharge. In the present study, “cut-down” was implemented when subjects regained stable control of the knee joint and were able to reproduce gait, which had been learned in unified AGT with a KAFO, during walking with an AFO.

#### 2.2.3. Group Interventions

Inpatients at the Kita-Fukushima Medical Center typically receive one hour of individualized physiotherapy per day, seven days per week. They receive conventional rehabilitation that included physical therapy (PT), occupational therapy (OT), and speech therapy (ST). All 45 patients received this conventional rehabilitation for 2–3 h every day. In cases where the general condition of the patient was unstable, PT was started from bedside and included a range of motion, sitting, and standing exercises. If permitted by the patient’s general condition, exercises for transfer maneuvers such as walking, mat work (e.g., getting up from the floor, which is common in Japanese everyday life), and stair climbing were performed incrementally, and the patients received individual PT for approximately 1 h per day. OT included activity of daily living (ADL, referring to daily self-care activities) training and arm exercises. ST included exercises for dysphagia and aphasia.

#### 2.2.4. Intervention of Unified AGT Subjects

All unified AGT subjects received unified AGT with a modular-type KAFO (Figure 2a). A ring lock hinge knee joint is used for the GI to hinge the ankle brace with the oil damper [18,19] to resist plantar flexion motion. Therefore, the hinged oil damper can resist plantarflexion during swing and adequate plantarflexion after heel contact [18,19,20]. The loading response period is therefore properly constructed because the first rocker function is maintained [18,19,20]. Because the hinged oil damper does not hinder ankle dorsiflexion, using GI promotes an inverted pendulum gait pattern (Figure 2b) known as a normal walking pattern [18,19,20]. We prepared six GIs (left and right, in S, M, and L sizes) to cover all patients. 

Unified AGT was provided by a physical therapist (Figure 2c); if it could not be performed easily, even if the therapist provided external assistance, step training was started for each of the affected and unaffected lower limbs. In these cases, AGT began after the step training.

After resolution of the patient’s knee instability, AFOs were given. Before the transition from a KAFO to an AFO, we confirmed that the AGT subjects were able to walk with an alternate gait pattern using their AFOs.

### 2.3. Clinical Measurements

The primary outcome was a change in FIM gait score at approximately 1, 2, or 3 months after admission. Once a month, after admission, an attending nurse evaluated the walking ability of all patients using FIM [13] gait score. FIM was developed to ensure uniformity in assessing ADL. FIM is an assessment tool for daily self-care activities and was developed to ensure uniformity in assessing ADL. Moreover, FIM has acceptable levels of validity and reliability when used for stroke as a measure of disability [14]. The FIM gait score ranges from 1 (requires assistance from two or more people) to 7 (independent). The severity of neurological motor impairments was evaluated with SIAS [15]. SAIS is an assessment set of stroke-related impairment, such as those in motor function, sensory function, muscle strength of the trunk and unaffected side, and higher brain function, as well as spasticity among others. SIAS is a comprehensive instrument for assessing stroke impairment, with well-established psychometric properties. The motor score for each item ranges from 0 (severely impaired) to 5 (normal), and the knee extension strength score on the unaffected side ranges from 0 (severely impaired) to 3 (normal) [15,16].

Patient motivation was evaluated using the VI at admission [14]. VI is an assessment tool for motivation for daily self-care activities, communication, and rehabilitation. It is composed of five subscales relating to common basic activities and has good validity and reliability. Each subscale has 0–2 points, and a higher value indicates good vitality [17].

Additionally, demographic information (age, sex, etiology, medical history of stroke, time from stroke onset to hospitalization in the convalescent ward, LOS, and duration of PT and OT treatment time) was reviewed.

### 2.4. Statistical Analyses

The demographic data were summarized using descriptive statistics. For comparisons of data between the groups, an unpaired *t*-test or Mann–Whitney U-test was used according to the results of the Shapiro–Wilk test. Furthermore, an χ2 test was used to compare the basic characteristics and nominal data. Improvements in gait ability were analyzed with analysis of variance (ANOVA) for split-plot factorial design (split-plot ANOVA) with training type (unified AGT or traditional) and FIM gait scores at four time points (initial, at 1, 2, and 3 months) as the test factors. A split-plot ANOVA was corrected using the Greenhouse–Geisser correction to account for the sphericity violation. Significant main effects pertaining to the type of training, the time course, and interaction were further explored with multiple comparisons using a Bonferroni correction, and “time from stroke to admission” was entered as a covariance. In addition, to clearly demonstrate early improvement, time taken to recover was analyzed using the Kaplan–Meier method with a log-rank test. The initial FIM gait score of all subjects was 1; therefore, in the present study, we defined recovery as the patient having reached a FIM gait score of 4 (minimum external assistance required) until the third FIM assessment. The alpha level was set at 0.05 for all analyses with IBM SPSS Statistics for the Mac (Version 25.0; IBM Corporation, Armonk, NY, USA).

## 3. Results

### 3.1. Comparison of Demographics Data

Table 1 shows the initial descriptive statistics, and the results of the χ2 test, unpaired t-test, and Mann–Whitney U-test comparisons. There were no differences between the AGT and HC subjects.

### 3.2. Effects of AGT on FIM Gait Scores

The split-plot ANOVA revealed significant changes in the FIM gait score over the time course (F [1.81, 77.83] = 29.11, *p* < 0.0001). However, the difference in FIM gait score between the type of training was not significant (F [1.43] = 0.504, *p* = 0.481). Moreover, there was no interaction effect between the type of training and the time course (F [1.81, 77.83] = 0.550, *p* = 0.562).

A multiple comparison analysis demonstrated a significant increase in the FIM gait score compared with its initial score (mean ± standard error, [95% CI], 1.00 ± 0.00, [1.00–1.00]), in the AGT subjects at 1, 2, and 3 months (1.73 ± 0.24 [1.26–2.21], *p* = 0.026; 2.12 ± 0.30 [1.50–2.74], *p* < 0.001; and 2.70 ± 0.41 [1.83–3.48], *p* < 0.001). However, in the HC subjects, a significant increase in the FIM gait score was observed compared with its initial score (1.00 ± 0.00, [1.00–1.00]) at 2 and 3 months (1.33 ± 0.17 [0.99–1.67], *p* = 0.34, 2.07 ± 0.22 [1.64–2.51], *p* < 0.001; and 2.40 ± 0.28 [1.84–2.98], *p* < 0.001, respectively) (Figure 3). 

The results of the survival analysis are shown in Figure 4. Three of the 15 AGT subjects and three out of the 30 HC subjects reached an FIM score of 4 within 3 months of admission. The log-rank test showed a significant difference between the unified AGT and HC groups (*p* = 0.02). 

In the present study, all AGT subjects successfully changed their KAFOs to AFOs (cut-down) and regained their gait function.

## 4. Discussion

In the current study, we compared the time course of gait function improvement between the AGT and HC subjects, case-matched according to severity of paresis, age, initial gait ability, motivation, and LOS. The HC subjects were provided with the same PT, OT, and ST treatment as those provided to the AGT subjects. However, our results indicate that the subjects who received AGT using GI yielded earlier improvements in their FIM gait scores from the first assessment, when compared with the HC subjects. In addition, survival analysis indicated that the AGT subjects reached an FIM score of 4 points quicker than the HC subjects. Thus, the AGT subjects achieved earlier improvement than the HC subjects in gait independence.

Intensive rehabilitation in a convalescent rehabilitation ward can be helpful in decreasing early impairments in ADLs, such as gait, in stroke patients with hemiplegia [4]. Because gait ability at discharge is a major concern when patients with disabilities return home, improving severe gait disability as early as possible is very important. Using an orthosis is effective for patients with severe hemiplegia because it allows them to gain stability in the affected foot and walk with reasonably good dynamic balance. In Japan, KAFOs have been used for stroke patients with hemiplegia for a long time [6]. Maeshima et al. [5] reported that patients with severe hemiplegia who undergo intensive standing and walking training with a KAFO in the early phase of hospitalization can return home sooner, thanks to an improvement in their capabilities to perform daily activities. Our results were comparable to those of previous studies [4,5] that reported using KAFOs in patients with severe hemiplegia is beneficial for improving gait ability and should be actively introduced during early and convalescent rehabilitation. In gait training, KAFOs are used for those with severe motor paralysis. Once stable control of the knee joint has been achieved, gait training is advanced with AFO, leading to an increased ability to walk independently [5]. The present study showed that, for stroke patients with severe hemiplegia, adopting a gait training scheme with KAFOs is a reasonable treatment approach. We consider that the benefits of AGT with a KAFO should be known by physicians and therapists.

In this study, all unified AGT subjects who were unable to walk without external assistance were provided with unified AGT with human assistance using GI. The AGT creates bilateral hip flexion–extension proprioceptive input. In contrast, the HC subjects were provided with our traditional gait training program. In our traditional gait training program, based on the subjective decision of each physiotherapist, even in cases with knee instability, the training was performed with AFOs instead of KAFOs. Therefore, we suspect that a difference in the timing of gait function improvement was created between the two groups as the input of information of hip extension under load [11,12] may induce increased muscle activity in the lower limb of the affected side in patients with severe hemiplegia. In addition, we speculate that the muscle activity of the lower limbs necessary for gait is enhanced by continuously providing unified AGT, contributing to the improvement of gait function in such patients. 

There are some limitations to the current study. The AGT was provided based on the evidence from previous studies of people with spinal cord injury [11,12], but the muscle activities of the lower limbs were not investigated in the present study. Additionally, the number of tested subjects was insufficient. Given that this is a HC study, the evidence may not be sufficient. Moreover, it might be assumed that a time bias exists because the HC group was evaluated prior to AGT becoming the standard of care. Hence, although there was no difference in the duration of PT and OT between the AGT and HC subjects, the differences in the duration and type of gait training could not be investigated enough due to the historical study design. A further prospective clinical trial is necessary to clarify the effect of AGT with the use of a KAFO with a larger number of stroke patients with severe hemiplegia, including the evaluation of the electromyographic signals of the lower limbs.

## 5. Conclusions

We performed a case-matched HC study to investigate whether unified AGT improved gait function quicker than our traditional gait training in hemiplegic stroke patients. The FIM gait score showed that the patients who underwent unified AGT with a KAFO achieved improved gait function quicker than those who underwent traditional gait training. Our findings suggest that unified AGT using a KAFO facilitates gait function recovery in patients with severe hemiplegia and knee instability.

## Figures and Tables

**Figure 1 brainsci-11-01430-f001:**
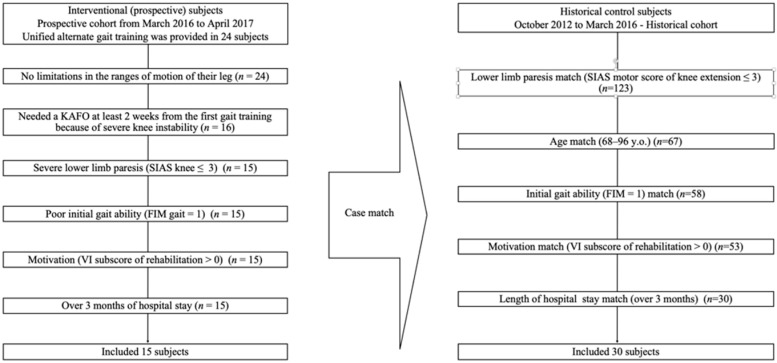
Subject enrollment flowchart.

**Figure 2 brainsci-11-01430-f002:**
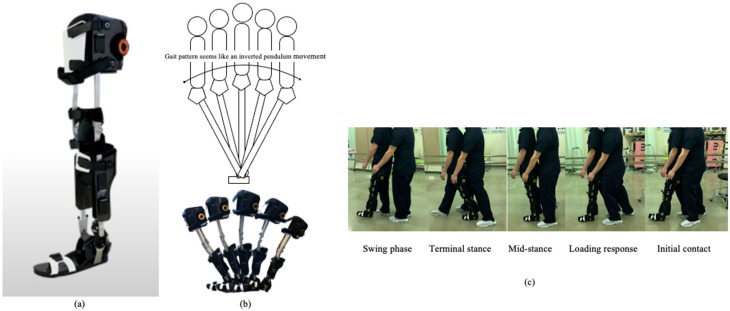
Gait Innovation and alternate gait training: (**a**) Gait Innovation. Gait Innovation has the following features: (1) a center fix system, which can adjust the circumference of the thigh cuff by turning the dial in the cuff; (2) a height adjustment system, which can adjust the height of the lower leg cuff, knee joint, and thigh cuff. A KAFO can also be easily changed to AFO by using a lever and slides; (3) a “foot holding mechanism,” where the heel is held in the orthosis via tightened ankle belts; (4) orthosis care, where the materials in a cuff or a foot component are wipeable and sterilizable with sterilization chemicals. (**b**) Gait Innovation can achieve inverted pendulum movements. (**c**) Alternate gait was created by using Gait Innovation and external assistance.

**Figure 3 brainsci-11-01430-f003:**
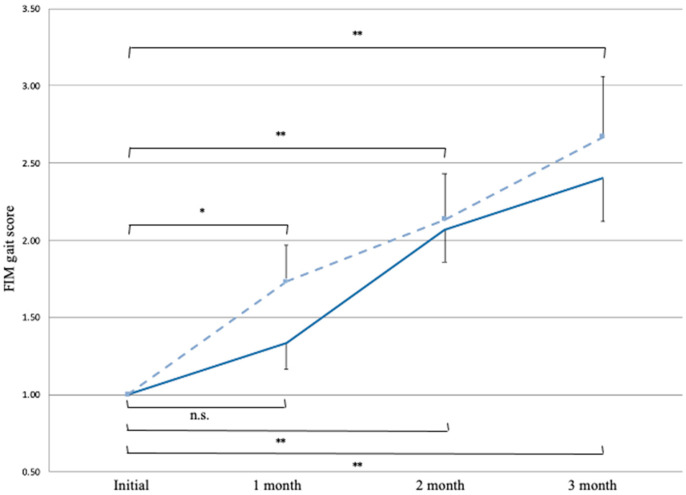
Multiple comparison using Bonferroni correction with covariance of “time from stroke to admission”. There was a significant increase in the FIM gait score from the initial measurement to the measurement at 1, 2, and 3 months in the AGT subjects; however, in the HC subjects, a significant increase was observed at 2 and 3 months. The dotted line indicates the AGT group, and the straight line indicates the HC group. Each point represents the mean. The vertical bars indicate standard errors. *, *p* < 0.05; **, *p* < 0.01; n.s., not significant; FIM, functional independence measure; AGT, alternate gait training; HC, historical control.

**Figure 4 brainsci-11-01430-f004:**
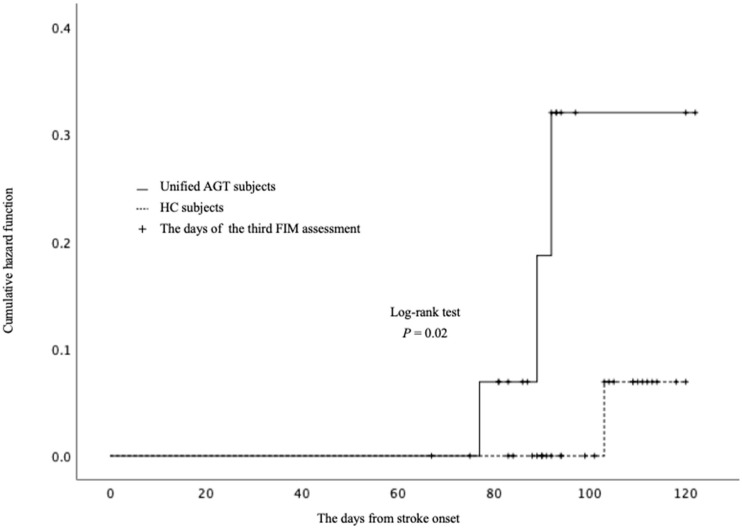
Survival analysis for reaching FIM gait score of 4. The vertical axis represents the cumulative hazard function of reaching an FIM gait score of 4 points. The horizontal axis represents the days from stroke onset. Solid line: AGT subjects; dashed line: HC subjects. Survival analysis indicated that the AGT subjects reached an FIM gait score of 4 quicker than the HC subjects during the FIM assessments period (nearly 12 weeks) following admission.

**Table 1 brainsci-11-01430-t001:** Comparison of characteristics of alternate gait training subjects and historical control subjects.

	Unified AGT Subjects			Historical Control Subjects			
	Mean ± SD or Number	Median, Interquartile Range	95% CI	Mean ± SD or Number	Median, Interquartile Range	95% CI	*p*-Value
Number of subjects	15			30			
Age†	77.1 ± 9.3	77.0, 14	71.9–82.2	77.9 ± 5.2	78.5, 7	75.9–79.8	0.76
Sex (male/female) ‡	5/10			14/16			0.39
Etiology (SAH/INF/ICH/INF and ICH) ‡	0/9/6/0			2/17/10/1			0.65
Paretic side (Right/Left) ‡	9/6			12/18			0.21
Stroke recurrence (yes/no) ‡	3/12			7/23			0.8
Time from stroke to admission (days) *	28.8 ± 11.2	27.0, 11	22.6–35.0	36.6 ± 14.6	37.0, 28	31.1–42.0	0.18
Length of hospital stay (days) †	120.5 ± 32.4	139.0, 46	102.5–138.4	120.5 ± 22.7	116.0, 35	112.1–129.0	0.99
Duration of the use of Gait Innovation (days)	41.8 ± 18.3	43.0, 28		none			
Duration of physical therapy (minutes) †	7980.0 ± 2388.0	6940.0, 4340	6658.0–9302.0	8330.0 ± 2642.0	8260.0, 3840	7343.8–9317.4	0.67
Duration of occupational therapy (minutes) †	6073.4 ± 2109.3	5240.0, 5640	4905.2–7241.4	6526.0 ± 1815.9	6700.0, 2000	5848.0–7204.0	0.61
Initial ADL (FIM total) *	36.9 ± 11.7	36.0, 19	30.4–43.3	33.9 ± 11.6	33.5, 17	29.6–38.4	0.43
Initial gait ability (FIM gait) †	1.0 ± 0.0	1, 0	1.0–1.0	1.0 ± 0.0	1, 0	1.0–1.0	All patients score is 1
Vitality index (E) motivation *	1.2 ± 0.4	1.0, 1	0.97–1.43	1.1 ± 0.4	1.0, 1	1.0–1.26	0.57
Vitality index total †	5.8 ± 1.8	5.0, 3	4.8–6.8	6.1 ± 1.9	6.0, 2	5.4 -6.8	0.71
SIAS motor hip *	1.2 ± 1.4	0.0, 3	0.4–2.0	1.3 ± 1.3	1.0, 2	0.8–1.8	0.84
SIAS motor knee *	1.2 ± 1.5	0.0, 3	0.4–2.0	1.1 ± 1.2	1.0, 2	0.7–1.5	0.88
SIAS motor ankle *	0.8 ± 1.3	0.0, 2	0.1–1.5	0.9 ± 1.4	0.0, 2	0.4–1.5	0.83
SIAS knee extension strength of the unaffected side *	2.0 ± 0.8	2.0, 2	1.6–2.4	1.8 ± 1.0	1.5, 2	1.37–2.2	0.48

CI: confidence interval; INF: cerebral infarction; ICH: intracranial hemorrhage; SAH: subarachnoid hemorrhage; FIM: functional independence measure; SIAS: Stroke Impairment Assessment Set; *: both groups were compared using the unpaired t-test; †: Mann–Whitney U-test; ‡: χ2 test.

## Data Availability

The datasets used and/or analyzed during the current study are available from the corresponding author on reasonable request.

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
