# Peer review of "Impact of Alternate Gait Training Using Knee–Ankle–Foot Orthoses with Oil Damper Ankle Hinge in Patients with Subacute Severe Hemiplegia"

_brainsci, 2021, doi:10.3390/brainsci11111430_

Round 1

Reviewer 1 Report

Overall this research is very interesting and was conducted in a well-organized manner. The paper is well-written; a good read. However, there are major weak spots that either need a more elaborate clarification or a better justification of choices.

This paper investigated the impact of alternate gait training (AGT) for patients with severe hemiplegia using knee-ankle-foot orthoses (KAFOs) compared with the conventional gait training. However, the authors showed that the patients who underwent unified AGT with a  KAFO achieved improved gait function more quickly than those who underwent conventional gait training.    

It would be great if authors might think about improve the introduction, e.g. KAFOS, AFOS, standard gait training not well introduced. in addition authors should elaborate on using knee instability since it is not clear what the authors meant by instability in many sections. Best recommendations I could suggest to make it clear for the readers early in the manuscript. 

In the study design section mentioning months not necessary I recommend to rephrase this section accordingly. 

Line 101 15 subjects instead of fifteen subjects. 
Line 121 e.g. instead of for example 
Line 130-135 should be in the acknowledgements!

Figure 1 better presentation would it be nice, since I could see red underlines!

Line 135-144 seems to be repeated in 160-168 for description of Gait innovation, so I suggest combine both paragraphs and make more consistence within the manuscript.  

SIAS mentioned but never introduce 
ADL mentioned but never introduce 
FIM mentioned but never introduce 
VI mentioned but never introduce 

For statistical analysis authors used Kaplan-Meier method! could authors explain why?

Line 211 degree of freedom not necessary!

Figure 3 legend is not exist, because seems better to understand without refer to the text! 

Line 248 horizontal instead of horizon!
Line 276 independently instead of independence!

Discussion seems to provide some evidence about using AGT instead of traditional gait training but never say why or provide a good arguments to support this hypothesis! Also authors could provide video about the training to accompany with the manuscript! and I totally recommend providing such video!

Author Response

Overall this research is very interesting and was conducted in a well-organized manner. The paper is well-written; a good read. However, there are major weak spots that either need a more elaborate clarification or a better justification of choices.

This paper investigated the impact of alternate gait training (AGT) for patients with severe hemiplegia using knee-ankle-foot orthoses (KAFOs) compared with the conventional gait training. However, the authors showed that the patients who underwent unified AGT with a KAFO achieved improved gait function more quickly than those who underwent conventional gait training.    

→Thank you for your peer review and comments that helped us to improve our manuscript. Please find our responses to each of your comments below.

It would be great if authors might think about improve the introduction, e.g. KAFOS, AFOS, standard gait training not well introduced.

in addition authors should elaborate on using knee instability since it is not clear what the authors meant by instability in many sections.

Best recommendations I could suggest to make it clear for the readers early in the manuscript.

→We have added the Video 1 and 2 to introduce AFOs, KAFOs, and AGT, as well as to show knee instability. Standard gait training is shown in the Video 1; however, we have changed the term “standard gait training” to “our traditional gait training” as it is difficult to clearly define the standard method, which was pointed out by Reviewer 2. The definition of "our traditional gait training" is described in Line 74-78: In our traditional gait training program, based on the subjective decision of each physiotherapist, even in cases with knee instability, the training was performed with AFOs instead of KAFOs.

In the study design section mentioning months not necessary

I recommend to rephrase this section accordingly.

→We have deleted "March" in Line 82 according to your comment.

Line 101 15 subjects instead of fifteen subjects.

Line 121 e.g. instead of for example

→We have revised the sentences according to your comment. (Line 110 and 130)

Line 130-135 should be in the acknowledgements!

→We have moved these sentences to the acknowledgements section in Line 323-326.

Figure 1 better presentation would it be nice, since I could see red underlines!

→We are sorry for this, and Figure 1 has been revised.

Line 135-144 seems to be repeated in 160-168 for description of Gait innovation, so I suggest combine both paragraphs and make more consistence within the manuscript. 

→As you pointed out, the same description was given twice. Therefore, we have deleted L135-144 (in old manuscript).

SIAS mentioned but never introduce

→We have added a sentence that explains SIAS as " SAIS is an assessment set of stroke related impairment such as that in motor function, sensory function, muscle strength of the trunk and unaffected side, and higher brain function, as well as spasticity among others" in Line 169-172.

ADL mentioned but never introduce

→We have added explanation as "ADL, referring to daily self-care activities." in Line 133.

FIM mentioned but never introduce

→We added" FIM is an assessment tool for daily self-care activities and was developed to ensure uniformity in assessing ADL." in Line 165-166.

VI mentioned but never introduce

→We added" VI is an assessment tool for motivation for daily self-care activities, communication, and rehabilitation." in Line 177-178.

For statistical analysis authors used Kaplan-Meier method! could authors explain why?

→To demonstrate early improvement clearly, we used the Kaplan-Meyer method, referring to the design of previous studies. In the text, we have revised the sentence as "In addition, to demonstrate early improvement clearly, time taken to recovery was analyzed using the Kaplan-Meier method with a log-rank test." in Line 194-196.

Line 211 degree of freedom not necessary!

→We have deleted "degree of freedom” in Line 208.

Figure 3 legend is not exist, because seems better to understand without refer to the text!

→We agree with you and have revised the Figure 3 legend, accordingly.  

horizontal instead of horizon!

→We are sorry for this error and have corrected accordingly. (Line 242)

Line 276 independently instead of independence!

→We are sorry for this error and have made a correction. (Line 272)

Discussion seems to provide some evidence about using AGT instead of traditional gait training but never say why or provide a good arguments to support this hypothesis! Also authors could provide video about the training to accompany with the manuscript! and I totally recommend providing such video!

→We have revised the sentence according to your suggestion in discussion section in Line 276-287, “In this study, all unified AGT subjects who were unable to walk without external assistance were provided with unified AGT with human assistance using GI.  The AGT creates bilateral hip flexion-extension proprioceptive input. In contrast, the HC subjects were provided with our traditional gait training program. In our traditional gait training program, based on the subjective decision of each physiotherapist, even in cases with knee instability, the training was performed with AFOs instead of KAFOs. Therefore, we suspect that a difference in the timing of gait function improvement was created between the two groups as the input of information of hip extension under load [11,12] may induce increased muscle activity in the lower limb of the affected side in patients with severe hemiplegia. In addition, we speculate that the muscle activity of the lower limbs necessary for gait is enhanced by continuously providing unified AGT, contributing to the improvement of gait function in such patients.

As mentioned previously, we have added the Video 1 and 2 according to your comment.

Reviewer 2 Report

Congratulations on this research. In this work, the impact of alternate gait training using knee-ankle-foot orthoses with oil damper was investigated in patients with subacute severe hemiplegia. The manuscript is well written and clearly structured. The method is well explained, and the data analysis approach is very good. The manuscript can be accepted after a minor review.

Abstract

“The results revealed that the unified AGT subjects showed earlier improvement than the HC subjects.” Please summarize more extensively the main results of your study and mention if the earlier improvement in the AGT subjects was statistically significant.

Introduction

“The purpose of the present study was to investigate whether AGT improves gait function more quickly than standard methods in hemiplegic stroke patients”. Please be more specific to the reader and describe in the introduction section which are the standard methods used to improve gait function in hemiplegic stroke patients. Moreover, the rationale of your study could be improved and extended. Please explain more clearly why you speculated that gait training using a KAFO would facilitate gait function recovery in patients with severe hemiparesis.

Materials and Methods

Lines 71-82: The text could be combined in one paragraph.

Line 92: “but was unable”. Please replace “was” with “were”.

Lines 147-148: “Because the hinged oil damper does not hinder ankle dorsiflexion, using GI promote an inverted pendulum gait pattern.” Please add a full stop before “because” and replace “promote” with “promotes”.

Lines 160-168: The text included in these lines should be moved to the legend of Figure 2.

Line 239: “Thus, the AGT subjects achieved earlier improvement than the HC subjects in gait

independence.” This sentence should be removed from the results sections and added to the discussion.

Lines 241-244: The text included in these lines should be moved to the legend of Figure 3.

Lines 247-251: The text included in these lines should be moved to the legend of Figure 4.

Discussion

Line 267: Reference [6] should be moved before the full stop.

Line 269: “created early during”. I propose the word “created” to be replaced with another more appropriate verb.

Lines 285-286: “Therefore, we assume that the AGT did not perform in HC subjects” Please revise this sentence.

Lines 288-289: “We assumed an induced muscle activity associate the quickly improvement of gait function in unified AGT subjects”. Please revise the sentence and replace the word “associate” with another more appropriate verb.

Author Response

Comments and Suggestions for Authors 2

Congratulations on this research. In this work, the impact of alternate gait training using knee-ankle-foot orthoses with oil damper was investigated in patients with subacute severe hemiplegia. The manuscript is well written and clearly structured. The method is well explained, and the data analysis approach is very good. The manuscript can be accepted after a minor review.

→We appreciate your kind review and suggestions, according to which we revised our manuscript as follows.

Abstract

“The results revealed that the unified AGT subjects showed earlier improvement than the HC subjects.” Please summarize more extensively the main results of your study and mention if the earlier improvement in the AGT subjects was statistically significant.

→We have revised the results in the abstract section for more extensive explanation as “The multiple comparison revealed a significant increase (p < 0.05) in the FIM gait score compared with its initial score in the subjects with unified AGT. However, this improvement was not seen in the HC subjects. In addition, the survival analysis of time taken to recovery revealed a significant difference between the subjects with unified AGT and HC subjects (p < 0.05). in the abstract. (Line 28-31)

Introduction

“The purpose of the present study was to investigate whether AGT improves gait function more quickly than standard methods in hemiplegic stroke patients”. Please be more specific to the reader and describe in the introduction section which are the standard methods used to improve gait function in hemiplegic stroke patients.

We changed the term “standard gait training” to “our traditional gait training” as it is what we actually meant.

Moreover, the rationale of your study could be improved and extended. Please explain more clearly why you speculated that gait training using a KAFO would facilitate gait function recovery in patients with severe hemiparesis.

→We have added “We hypothesized that the provision of afferent information from the bilateral hip joints by unified AGT would induce locomotor-patterned leg muscle activation and improve gait function early in hemiplegic stroke patients. (see Video 2)” in Line 78-80.

Materials and Methods

Lines 71-82: The text could be combined in one paragraph.

→We have combined the paragraphs according to the suggestion. (Line 82-88)

Line 92: “but was unable”. Please replace “was” with “were”.

→We are sorry for this error and have corrected accordingly. (Line 101)

Lines 147-148: “Because the hinged oil damper does not hinder ankle dorsiflexion, using GI promote an inverted pendulum gait pattern.” Please add a full stop before “because” and replace “promote” with “promotes”.

→We are sorry for these errors and have made corrections. (Line 142-143)

Lines 160-168: The text included in these lines should be moved to the legend of Figure 2.

→The same suggestion was given from Reviewer 1, and we have moved these sentences to the legend of Figure 2.

Line 239: “Thus, the AGT subjects achieved earlier improvement than the HC subjects in gait independence.” This sentence should be removed from the results sections and added to the discussion.

→We have removed this sentence from the results section and added to the discussion section in Line 255-256.

Lines 241-244: The text included in these lines should be moved to the legend of Figure 3.

→We have revised the legend of Figure 3, according to your comment.

Lines 247-251: The text included in these lines should be moved to the legend of Figure 4.

→We have revised the legend of Figure 4, according to your comment.

Discussion

Line 267: Reference [6] should be moved before the full stop.

→We have revised the text, according to your comment. (Line 263)

Line 269: “created early during”. I propose the word “created” to be replaced with another more appropriate verb.

→We have revised this sentence as" Maeshima et al. [5] reported that patients with severe hemiplegia who undergo intensive standing and walking training with a KAFO in the early phase of hospitalization can return home sooner, thanks to an improvement in their capabilities to perform daily activities.” (Line 263-266)

Lines 285-286: “Therefore, we assume that the AGT did not perform in HC subjects” Please revise this sentence.

→We have revised this sentence as “Therefore, we suspect that a difference in the timing of gait function improvement was created between the two groups as the input of information of hip extension under load [11,12] may induce increased muscle activity in the lower limb of the affected side in patients with severe hemiplegia.”. (Line 281-285)

Lines 288-289: “We assumed an induced muscle activity associate the quickly improvement of gait function in unified AGT subjects”. Please revise the sentence and replace the word “associate” with another more appropriate verb.

→We have revised this sentence as " In addition, we speculate that the muscle activity of the lower limbs necessary for gait is enhanced by continuously providing unified AGT, contributing to the improvement of gait function in such patients". (Line 285-287)